# Deep Learning-Based Classification of Uterine Cervical and Endometrial Cancer Subtypes from Whole-Slide Histopathology Images

**DOI:** 10.3390/diagnostics12112623

**Published:** 2022-10-28

**Authors:** JaeYen Song, Soyoung Im, Sung Hak Lee, Hyun-Jong Jang

**Affiliations:** 1Department of Obstetrics and Gynecology, Seoul St. Mary’s Hospital, College of Medicine, The Catholic University of Korea, Seoul 06591, Korea; 2Department of Hospital Pathology, St. Vincent’s Hospital, College of Medicine, The Catholic University of Korea, Seoul 16247, Korea; 3Department of Hospital Pathology, Seoul St. Mary’s Hospital, College of Medicine, The Catholic University of Korea, Seoul 06591, Korea; 4Catholic Big Data Integration Center, Department of Physiology, College of Medicine, The Catholic University of Korea, Seoul 06591, Korea

**Keywords:** computational pathology, computer-aided diagnosis, convolutional neural network, digital pathology

## Abstract

Uterine cervical and endometrial cancers have different subtypes with different clinical outcomes. Therefore, cancer subtyping is essential for proper treatment decisions. Furthermore, an endometrial and endocervical origin for an adenocarcinoma should also be distinguished. Although the discrimination can be helped with various immunohistochemical markers, there is no definitive marker. Therefore, we tested the feasibility of deep learning (DL)-based classification for the subtypes of cervical and endometrial cancers and the site of origin of adenocarcinomas from whole slide images (WSIs) of tissue slides. WSIs were split into 360 × 360-pixel image patches at 20× magnification for classification. Then, the average of patch classification results was used for the final classification. The area under the receiver operating characteristic curves (AUROCs) for the cervical and endometrial cancer classifiers were 0.977 and 0.944, respectively. The classifier for the origin of an adenocarcinoma yielded an AUROC of 0.939. These results clearly demonstrated the feasibility of DL-based classifiers for the discrimination of cancers from the cervix and uterus. We expect that the performance of the classifiers will be much enhanced with an accumulation of WSI data. Then, the information from the classifiers can be integrated with other data for more precise discrimination of cervical and endometrial cancers.

## 1. Introduction

Uterine cervical and endometrial cancers are two major cancer types threatening women’s health worldwide [1]. Although they originate from the same organ, i.e., uterus, cervical and endometrial cancers have different subtypes with different clinical outcomes [2,3,4,5,6]. The main histologic subtypes of cervical cancers are squamous cell carcinoma and endocervical adenocarcinoma. The two major histologic subtypes of endometrial cancers are endometrioid adenocarcinoma and serous adenocarcinoma. Because management and prognosis are different between the subtypes, differential diagnosis is crucial for proper treatment decisions. Furthermore, an endometrial and endocervical origin for an adenocarcinoma should be distinguished considering the marked differences in their management [7]. The first step for the discrimination of the subtypes of these cancers is to investigate hematoxylin and eosin (H&E)-stained tissue slides by pathologists. However, the visual discrimination of subtypes is not always clear because some morphologic features are overlapping [7,8]. Furthermore, there is considerable inter- and intra-observer variations in the histological subtyping by pathologists [8]. Although various immunohistochemical markers can help distinguish the subtypes, there is no definitive marker [7,8]. Therefore, ancillary methods for the discrimination of the subtypes of cervical and endometrial cancers, and also the origin of the cancers are necessary to improve treatment decisions.

Because whole-slide images (WSIs) were approved for primary diagnostic purposes, many pathologic laboratories have been adopting digitized diagnosis processes [9]. The digitization enabled computer-aided analysis of pathologic tissues. Computer-aided analysis of H&E-stained WSIs could provide valuable information in a cost- and time-effective manner, considering the wide availability of H&E-stained pathologic tissue slides for most cancer patients. Recently, deep learning (DL) has been widely applied for various analysis tasks on H&E-stained WSIs [10]. DL usually performs better than many previous machine learning methods because it can automatically learn the most discriminative features directly from large datasets [11]. Many studies showed that DL can correctly diagnose various cancers from WSIs [12]. Furthermore, DL can even detect molecular alterations of cancer tissues from H&E-stained WSIs [13]. Therefore, DL has tremendous potential to improve the precision of pathologic diagnosis with minimal additional cost.

In the present study, we applied sequential DL models for the subtyping of cervical and endometrial cancers. First, cervical and endometrial cancer regions were automatically selected with DL models. Then, two separate DL models were trained to discriminate cervical and endometrial cancers into cervical squamous cell carcinoma and endocervical adenocarcinoma, and into endometrioid endometrial adenocarcinoma and serous endometrial adenocarcinoma, respectively. Furthermore, we trained an additional DL model to discriminate whether an adenocarcinoma has an endocervical or endometrial origin. The three models showed excellent performance proving the potential of DL for the discrimination of subtypes in gynecologic tumors.

## 2. Materials and Methods

### 2.1. Datasets

Classifiers for the subtypes of cervical and endometrial cancers and the origin of adenocarcinomas were trained with the WSIs provided by The Cancer Genome Atlas (TCGA) program. From the TCGA cervical (TCGA-CESC) and endometrial (TCGA-UCEC) datasets, we collected formalin-fixed paraffin-embedded (FFPE) slides after the basic slide quality reviews. The TCGA-CESC dataset provided slides from 255 patients for cervical squamous cell carcinoma and from 47 patients for endocervical adenocarcinoma. From the TCGA-UCEC dataset, tissue slides of 399 and 109 patients were obtained for endometrioid endometrial adenocarcinoma and serous endometrial adenocarcinoma, respectively. When there are huge differences in the numbers of data between the classes, performance evaluation can be skewed by the majority class. Therefore, we randomly selected 70 and 160 patients for cervical squamous cell carcinoma and endometrioid endometrial adenocarcinoma, respectively, to make the differences between the major and minor classes under 1.5-fold.

The performance of the classifier for the subtypes of endometrial carcinoma was also evaluated on The National Cancer Institute’s Clinical Proteomic Tumor Analysis Consortium (CPTAC) endometrial cancer dataset (CPTAC-UCEC). There were 83 patients for endometrioid endometrial adenocarcinoma and 12 patients for serous endometrial adenocarcinoma.

### 2.2. Deep Learning Model

To fully automate the classification tasks, we sequentially applied different DL-based classifiers to the WSIs (Figure 1). The WSIs were divided into non-overlapping, 360 × 360-pixel image patches at 20× magnification because a WSI is too big to be analyzed by a current DL-system as a whole. In a WSI, various artifacts can exist including air bubbles, blurring, compression artifacts, pen markings, and tissue folding. Patches with these artifacts should be discarded because they can interfere with proper learning of relevant features. In our previous study for gastric cancer subtyping, we trained a simple DL classifier that can discriminate various artifacts and white backgrounds all at once [14]. The DL network consisted of three convolution layers with 12 [5 × 5] filters, 24 [5 × 5] filters and 24 [5 × 5] filters, each followed by a [2 × 2] max-pooling layer. We reused the classifier and only proper tissue image patches were selected for the next steps (Figure 1a).

Cancer subtype classifiers should be trained on the cancer tissues. Therefore, normal and tumor tissue classifiers are prerequisites for cancer subtyping. To train the normal/tumor classifiers, two pathologists (S.I. and S.H.L.) annotated normal and tumor regions for cervical and endometrial cancer tissue slides (Figure 2 left panels). Then, normal and tumor tissue image patches were collected based on the annotation. From these patches, classifiers to discriminate normal and tumor tissues for cervical and endometrial cancers were trained separately for each cancer type.

Next, we trained classifiers for the subtypes of cervical and endometrial cancers, and the origin of adenocarcinomas on prominent tumor tissue patches selected by the normal/tumor classifiers. To evaluate the general performance of the classifiers for the TCGA-CESC and -UCEC datasets, 5-fold cross validation was adopted. Therefore, the WSIs were split into 5 non-overlapping patient-level subsets and classifiers were trained and evaluated for each subset. As we noted, 70 and 160 patients for cervical squamous cell carcinoma and endometrioid endometrial adenocarcinoma were selected for evaluation. However, performance can be enhanced when the classifiers were exposed to more various tissue images during training. Therefore, we randomly sampled tumor image patches from all cervical squamous cell carcinoma and endometrioid endometrial adenocarcinoma WSIs other than the test sets to match the 1.5-fold data ratio of major/minor class tissue patches for training, as this strategy could include a greater variety of tissue images. Therefore, we included sampled data from all patients other than the test sets during training and selected patients for the testing to avoid skewed test results. For the selection of the samples, we made a random selection to avoid selection biases from human selectors. The numbers of image patches used for the training of the classifiers were summarized in Appendix A.

Inception-v3 model was adopted for the normal/tumor, cancer subtypes, and origin classifiers because the Inception-v3 model yielded good results for normal/tumor classification or tissue subtype classification in our previous studies [14,15]. The models were implemented using the Tensorflow deep learning library version 1.15 (http://tensorflow.org (accessed on 22 January 2022)). The overall structure of the model is presented in Appendix A. RMSPropOptimizer was adopted to optimize the model and the hyperparameters were as follows: initial learning rate 0.1, number of epochs per decay 10.0, learning rate decay factor 0.16, RMSPROP decay 0.9, RMSPROP_MOMENTUM 0.9, RMSPROP_EPSILON 1.0. Tissue images were color normalized before the training and testing. During training, data augmentation techniques such as random rotation by 90° and random horizontal/vertical flipping were applied to the tissue patches. Four computer systems equipped with an Intel Core i9-12900K Processor (Intel Corporation, Santa Clara, CA, USA) and dual NVIDIA RTX 3090 GPUs (NVIDIA corporation, Santa Clara, CA, USA) were used for the training and testing of the models.

### 2.3. Visualization and Statistics

To visualize the distribution of different tissue types, heatmaps of classification results of tissue patches were overlaid on the WSIs with specific colors demonstrated in Figure 1. To obtain the overall classification result of a WSI, patch classification results were averaged to obtain the result for the WSI. Receiver operating characteristic (ROC) curves and area under the curves for the ROC curves (AUROCs) were presented to demonstrate the performance of each classifier. For 5-fold cross validated datasets, ROC curves for the folds with the lowest and highest AUROCs and for the concatenated results of all 5 folds were provided for more precise evaluation of the performance of the classifiers. For the concatenated results of all 5 folds, 95% confidence intervals (CIs) were presented. To obtain accuracy, sensitivity, specificity and F1 score of the classification results, cutoff values yielding maximal Youden index (sensitivity + specificity − 1) were adopted.

When a comparison between the ROC curves is necessary, Venkatraman’s permutation test with 1000 iterations was applied [16]. A *p*-value < 0.05 was considered significant.

### 2.4. Ethical Statement

Informed consent of patients in the TCGA cohorts was acquired by the TCGA consortium [17]. The Institutional Review Board of the College of Medicine at The Catholic University of Korea approved this study (XC21ENDI0031K).

## 3. Results

### 3.1. Normal/Tumor Classification

To classify the subtypes of cancer tissues, proper cancer tissue image patches should be selected (Figure 1). First, we removed image patches containing various artifacts and white background with a tissue/non-tissue classifier from our previous study [14]. Then, normal/tumor classifiers for cervical and endometrial cancers were trained based on pathologists’ annotation (Figure 2). Pathologists annotated 100 slides for each cervical and endometrial cancer. The normal/tumor classifiers were trained with 80 slides and tested on the remaining 20 slides. The representative WSIs in Figure 2 are the cervical and endometrial cancer WSIs from the test sets. The classification results of the normal/tumor classifiers matched well with the pathologists’ annotation. The AUROCs for the patch-level classification results of the normal/tumor classifiers are 0.982 and 0.999 for cervical and endometrial cancers, respectively.

### 3.2. Cervical Cancer Subtypes Classification

With the tissue/non-tissue and normal/tumor classifiers, we can collect proper tumor patches for the training of the cancer subtype classifiers. First, we trained classifiers for the cervical cancer subtypes. The patches from a WSI are labeled as either cervical squamous cell carcinoma or endocervical adenocarcinoma based on the information obtained from cBioPortal for Cancer Genomics (https://www.cbioportal.org/ (accessed on 12 March 2022)). Then, separate classifiers were trained to distinguish the subtypes for each 5-fold. For each fold, four classifiers were trained repeatedly and a classifier yielding the best AUROC was used to present the results. The classification results of cervical squamous cell carcinoma and endocervical adenocarcinoma are presented in Figure 3. The upper panels show the representative WSIs of clear cervical squamous cell carcinoma, clear endocervical adenocarcinoma, and confusing case with mixed classification results. The ROC curves of slide-level classification results for folds with the lowest and highest AUROCs and concatenated results of all 5-folds are presented in the lower panels. The AUROCs were 0.979 and 1.000 for the folds with the lowest and highest AUROCs, respectively. The AUROC for the concatenated results was 0.977 (95% CI, 0.957–0.998).

### 3.3. Endometrial Cancer Subtypes Classification

Next, we trained other classifiers for the endometrial cancer subtypes. The patches from a WSI are labeled as either endometrioid endometrial adenocarcinoma or serous endometrial adenocarcinoma based on the information obtained also from the cBioPortal. The classification results are presented in Figure 4a. The representative WSIs of clear endometrioid endometrial adenocarcinoma, clear serous endometrial adenocarcinoma, and confusing case with mixed classification results are presented in the upper panels. The AUROCs were 0.923 and 0.982 for the folds with the lowest and highest AUROCs, respectively. The AUROC for the concatenated results was 0.944 (95% CI, 0.916–0.969).

It is of interest whether the classifiers trained on the TCGA datasets work well or not on other datasets. Therefore, we tested the classifier on the CPTAC-UCEC dataset. CPTAC-UCEC provides multiple WSIs for a patient with pure normal tissue WSIs (Figure 5a). We discarded normal WSIs and selected all WSIs with more than 30% of tumor tissue regions for the testing. The classification results are presented in Figure 4b. The AUROC was 0.826 (95% CI, 0.727–0.925), much poorer compared to the AUROC for the TCGA dataset (*p* < 0.05 between CPTAC and TCGA by Venkatraman’s permutation test).

### 3.4. Tumor Origin Classification

Lastly, we trained classifiers to distinguish the origin of adenocarcinomas: endocervical adenocarcinoma vs. endometrioid endometrial adenocarcinoma. The classification results are presented in Figure 6. The upper panels show the representative WSIs of clear endocervical adenocarcinoma, clear endometrioid endometrial adenocarcinoma, and confusing case with mixed classification results. The AUROCs were 0.904 and 0.987 for the folds with the lowest and highest AUROCs, respectively. The AUROC for the concatenated results was 0.939 (95% CI, 0.896–0.982).

In Table 1, accuracy, sensitivity, specificity, and F1 score of the classification results for these classifiers were presented with cutoff values yielding maximal Youden index (sensitivity + specificity − 1).

## 4. Discussion

In the present study, we investigated the feasibility of DL-based classification for the subtypes of cervical and endometrial cancers and the site of origin of adenocarcinomas. Although the performance of the classifiers was not perfect, high AUROCs of all the classifiers revealed the potential of DL-based classification of H&E-stained tissue slides of cervical and uterine cancers. The performance can be much enhanced when more WSI data can be collected for the training of the classifiers.

The DL-based classifiers for cervical cancer showed the best performance among the classifiers in the study. Pure adenocarcinoma and squamous cell carcinoma of the cervix can be relatively clearly separable because their morphologies have many differences [5]. However, there are also confusing cases including adenosquamous carcinoma which is defined as a tumor with both glandular and squamous components. This explains why the classifier could not accomplish perfection. In clinical practice, tissue slides with mixed classification results need more careful attention by pathologists when a DL-based assistant system for tissue slides is adopted.

Serous endometrial adenocarcinoma represents only about 10% of endometrial carcinomas. However, it is responsible for almost 40% of cancer deaths [8,18]. The distinction between endometrioid and serous endometrial adenocarcinoma is not very clear. Although serous carcinoma typically shows a predominant papillary growth pattern, which is also found in some endometrioid carcinomas. Antibodies for p53, p16, IMP2, and IMP3 can help to distinguish serous endometrial adenocarcinoma, but the markers are not definitive [19]. Therefore, there is an opportunity for DL-based classifiers to improve the diagnostic accuracy of subtypes of endometrial cancers.

One of the important issues of DL application is the generalizability of trained classifiers for external datasets. The TCGA-trained classifiers did not perform well on the CPTAC dataset in the present study. There can be various reasons for the decreased performance. First, the quality of H&E-stained tissue slides can vary between TCGA and CPTAC datasets because of the differences in tissue processing including tissue cutting, fixation, dye concentration, and staining time [20]. Furthermore, the differences in the settings of the slide scanners can also affect the color features of the WSIs. Although we normalized color, it may not be able to overcome the innate differences in the datasets. In addition, there are many other differences between TCGA and CPTAC datasets. CPTAC dataset contains not only FFPE tissues but also frozen tissue sections (Figure 5b). In our previous study, we clearly demonstrated that the classifiers trained on either frozen or FFPE tissue did not perform well on another tissue type [21]. Therefore, the classifiers trained on the TCGA-UCEC FFPE tissues cannot perform properly on the CPTAC frozen tissues. Furthermore, the CPTAC dataset also contains small tissue samples such as biopsy or small curettage specimens (Figure 5c). The dilatation and curettage may be able to deform tissue morphology. In addition, because biopsy samples have fundamental limitations in reflecting the overall contour of tumor histomorphology, the classifiers trained on resection specimens may not perform well on biopsy or small curettage tissues. Whatever the reason, the limited generalizability suggests that the TCGA dataset is not enough to train a classifier performing generally well on real-world problems. More data from various institutes should be collected to establish high generalizability. Recently, many countries started to construct large datasets of pathologic tissue slides [22,23]. Therefore, the performance and generalizability of DL-based tissue classifiers will be much enhanced with the accumulation of more training data in the near future.

The distinction of the site of origin between cervical adenocarcinomas and endometrial adenocarcinomas is important for clinical decisions especially for tumors involving both the endometrium and the endocervix or for tumors with multiple lesions [7]. The decision can be supported by immunohistochemistry for ER, p16, CEA, and vimentin or HPV in situ hybridization [5]. However, there is no decisive marker and additional methods are necessary to support the distinction. It is strongly recommended that various information including clinicopathologic, immunohistochemical, and molecular data should be integrated for proper differentiation of these cancers. We suggest that information from the DL-based classifier can also be integrated into these data for more accurate decisions.

In the present study, we applied DL to classify H&E-stained tissues of cervical and endometrial cancers. There have been other studies applying DL to assist the analysis of gynecologic tumors. Many studies tried to improve cervical cancer screening results based on cervical cytology tests [24,25,26]. In these studies, DL can discriminate normal/cancer cells from conventional Pap smear or liquid-based cytology. Grades of cervical intraepithelial neoplasia can be determined by DL from either colposcopy images [27,28] or histology images [29]. DL can also analyze hysteroscopy images to discriminate different types of endometrial legions [30,31]. Normal endometrium, endometrial polyp, endometrial hyperplasia, and endometrial adenocarcinoma can be discriminated by DL from H&E-stained histopathologic slides [32]. Molecular profiles such as molecular subtypes or microsatellite instability status of endometrial cancers can be predicted by DL directly from H&E-stained WSIs [33]. These studies indicate that DL has tremendous potential to support the assessment of patients with gynecologic tumors.

However, there are also limitations of DL. First, it is almost impossible for human interpreters to understand how DL reaches to the classification results. This “black-box” nature is one of the most important hurdles for the adoption of DL in clinical practice [34]. The effort to enhance the interpretability of DL is actively ongoing [35]. Second, DL cannot perform well in inexperienced settings although the difference is not tremendous. For example, a classifier trained on FFPE tissues has limited performance on frozen tissues although the difference is not limiting to human. Therefore, separate DL models should be trained for slightly different settings. Otherwise, a huge dataset covering every variation should be used to train a widely available model.

In the present study, we demonstrated the feasibility of DL-based classifiers for the subtypes of cervical and endometrial cancers and the site of origin of adenocarcinomas. Although there is still room for improvement, our results showed that DL can capture selective features for the discrimination of cancer tissues. We believe the performance will be much enhanced with an accumulation of training data in the near future. The classification results of DL can be integrated with other clinical information for a more precise analysis of cervical and endometrial cancers.

## Figures and Tables

**Figure 1 diagnostics-12-02623-f001:**
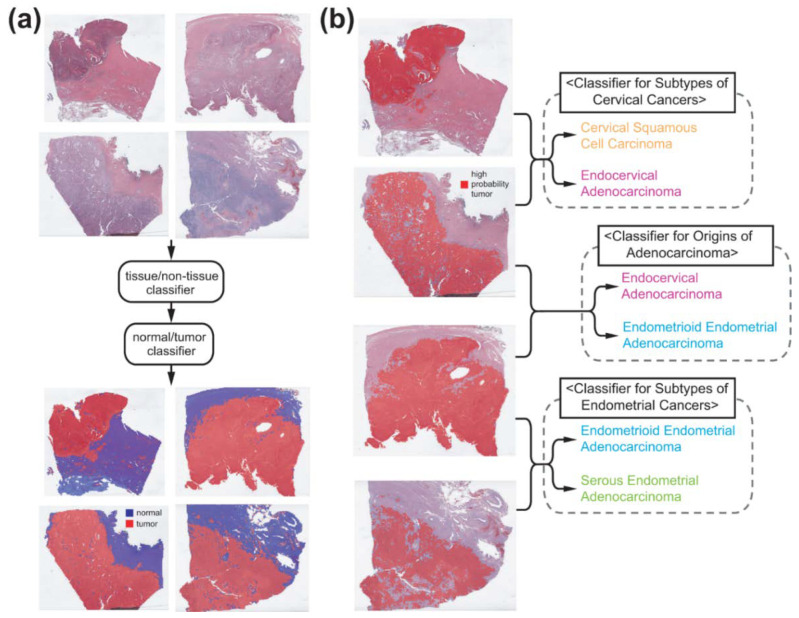
Classification procedure. (**a**) Sequential application of tissue/non-tissue and normal/tumor classifiers can discriminate proper tumor tissues. (**b**) Three separate classifiers for subtypes of cervical cancers, subtypes of endometrial cancers, and site of origin for adenocarcinomas were trained from tumor tissue image patches.

**Figure 2 diagnostics-12-02623-f002:**
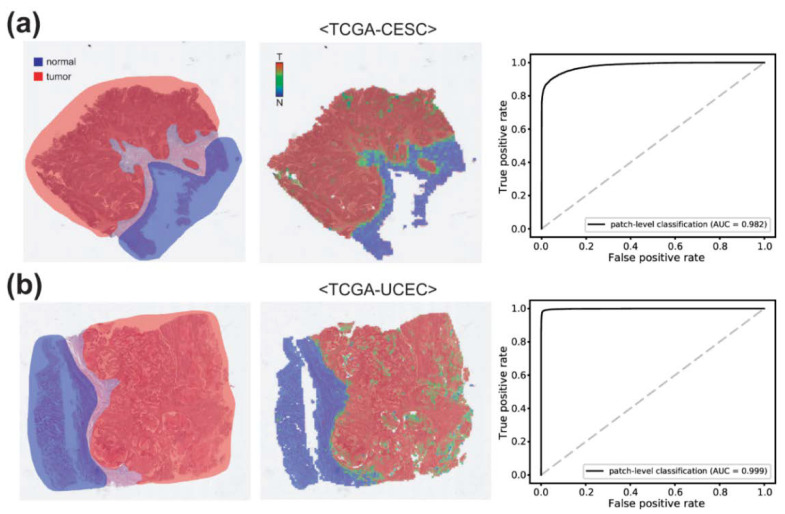
Normal/tumor classification results for (**a**) cervical and (**b**) endometrial cancers. Left panels: annotation made by pathologists. Middle panels: classification results of the normal/tumor classifiers. Right panels: Receiver operating characteristic curves for normal/tumor classification results. AUC: area under the curve.

**Figure 3 diagnostics-12-02623-f003:**
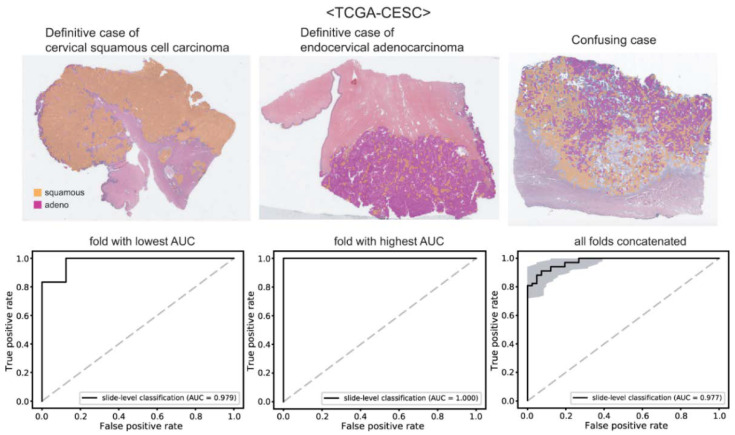
Classification results for cervical cancer subtypes. Upper panels: the representative whole slide images of clear cervical squamous cell carcinoma, clear endocervical adenocarcinoma, and confusing case with mixed classification results. Lower panels: the receiver operating characteristic curves of slide-level classification results for folds with the lowest and highest area under the curve (AUC) and concatenated results of all 5-folds.

**Figure 4 diagnostics-12-02623-f004:**
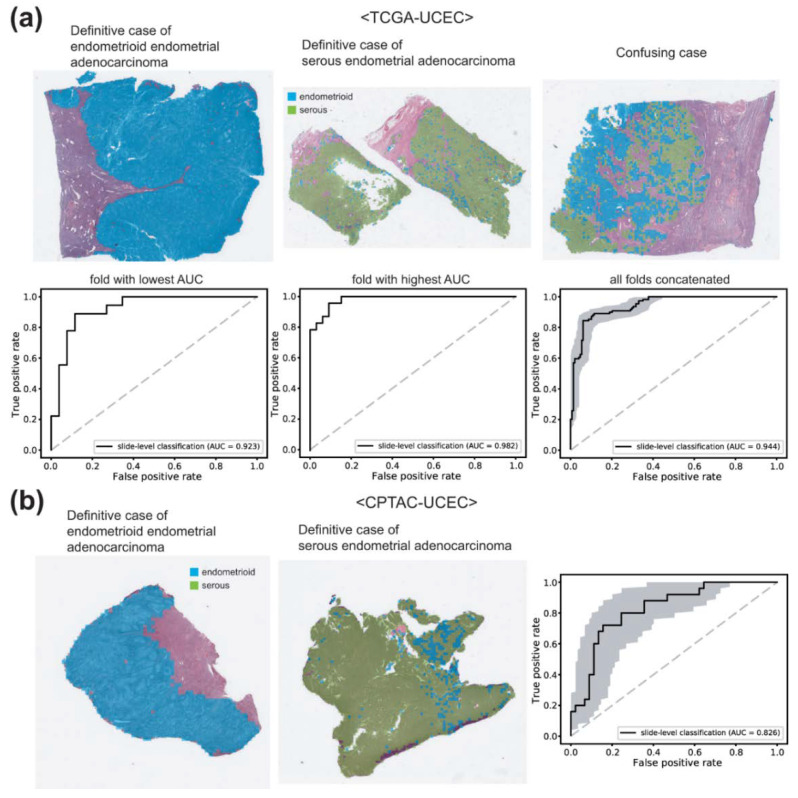
Classification results for endometrial cancer subtypes. (**a**) Results for the TCGA-UCEC dataset. Upper panels: the representative whole slide images (WSIs) of clear endometrioid endometrial adenocarcinoma, clear serous endometrial adenocarcinoma, and confusing case with mixed classification results. Lower panels: the receiver operating characteristic (ROC) curves of slide-level classification results for folds with the lowest and highest area under the curve (AUC) and concatenated results of all 5-folds. (**b**) The classification results of the CPTAC-UCEC dataset by the classifier trained with the TCGA-UCEC dataset. Left two representative WSIs demonstrate clear endometrioid endometrial adenocarcinoma and clear serous endometrial adenocarcinoma. The ROC curve is obtained from all CPTAC-UCEC tissues with more than 30% of tumor tissue regions.

**Figure 5 diagnostics-12-02623-f005:**
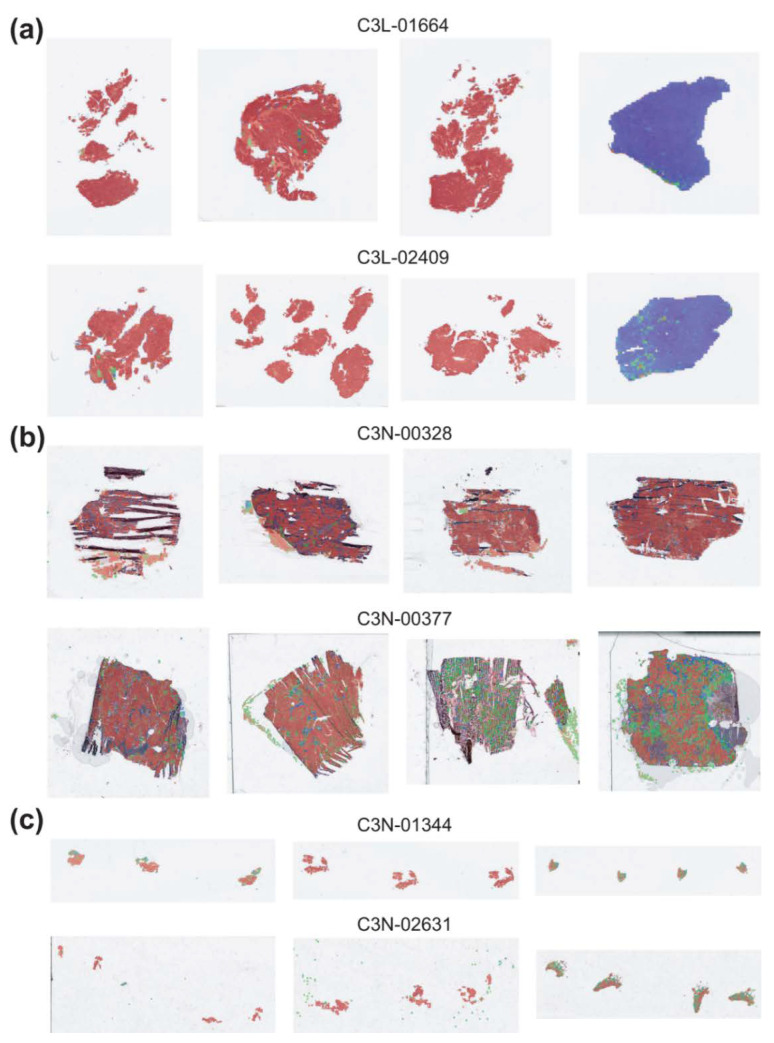
Characteristics of CPTAC-UCEC tissues. Examples of tissues from six patients indicated by IDs. (**a**) Patients with both pure tumor and pure normal tissue samples. (**b**) Patients with frozen tissue samples. (**c**) Patients with small curettage tissue samples.

**Figure 6 diagnostics-12-02623-f006:**
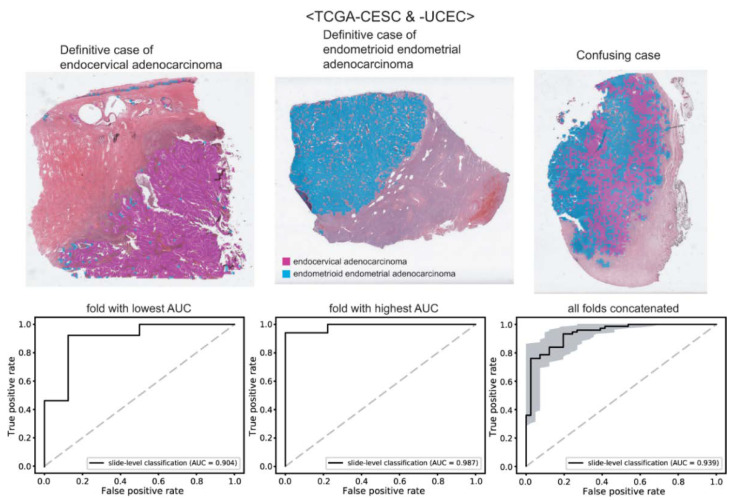
Classification results for the origin of adenocarcinomas. Upper panels: the representative whole slide images of clear endocervical adenocarcinoma, clear endometrioid endometrial adenocarcinoma, and confusing case with mixed classification results. Lower panels: the receiver operating characteristic curves of slide-level classification results for folds with the lowest and highest area under the curve (AUC) and concatenated results of all 5-folds.

**Table 1 diagnostics-12-02623-t001:** Accuracy, sensitivity, specificity, and F1 score of the classification results. The measures were obtained with cutoff values yielding maximal Youden index (sensitivity + specificity − 1).

	Accuracy	Sensitivity	Specificity	F1 Score
TCGA-CESC cervical squamous cell carcinoma/ endocervical adenocarcinoma	0.917	0.912	0.927	0.932
TCGA-UCEC endometrioid endometrial adenocarcinoma/ serous endometrial adenocarcinoma	0.899	0.846	0.939	0.876
CPTAC-UCEC endometrioid endometrial adenocarcinoma/ serous endometrial adenocarcinoma	0.757	0.8	0.733	0.702
TCGA-CESC/UCEC endocervical adenocarcinoma/ endometrioid endometrial adenocarcinoma	0.888	0.933	0.805	0.915

## Data Availability

The TCGA data presented in this study are openly available in the GDC data portal (https://portal.gdc.cancer.gov/ (accessed on 11 April 2022)). The CPTAC data presented in this study are openly available in The Cancer Imaging Archive website (http://www.cancerimagingarchive.net/ (accessed on 29 April 2022)). Further information is available from the corresponding authors upon request.

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
