# Peer review of "Deep Learning-Based Classification of Uterine Cervical and Endometrial Cancer Subtypes from Whole-Slide Histopathology Images"

_diagnostics, 2022, doi:10.3390/diagnostics12112623_

Round 1

Reviewer 1 Report

No comments.

Author Response

Because reviewer 1 had no comments, we revised our manuscript based on the comments from reviewers 2 and 3.

Reviewer 2 Report

The architecture of the DL should be elaborated in figures and paragraphs.

How many images are used for training and testing the DL?

Supplementary the figure and tables should be put in the main article.

Author Response

The architecture of the DL should be elaborated in figures and paragraphs.
-> We added a new Supplementary FigS1 to explain the structure of our DL model. Furthermore, the Materials and Methods section was modified for a more detailed explanation of the models.
(Line: 101-103)(Line: 136)

How many images are used for training and testing the DL?
-> We added a new Supplementary Table S1 to describe the number of image patches used for the training of the DL models.
(Line: 129-131)

Supplementary the figure and tables should be put in the main article.
-> We moved the original Supplementary Figure S1 and Supplementary Table S2 to the main manuscript. Supplementary Table S1 was moved to the Materials and Methods section as texts. 
(Line: 137-139)

Reviewer 3 Report

The first author serves in the Department of Obstetrics and Gynecology, Seoul St. Mary's Hospital, College of Medicine. Uterine cervical and endometrial cancers have different subtypes with different clinical outcomes. I think the authors should have enough experience in processing relevant medical images.

Approach: Deep learning-based classification.

Dataset: The TCGA-CESC dataset provided slides from 255 patients for cervical squamous cell carcinoma and from 47 patients for endocervical adenocarcinoma.

Results: The results of this paper showed a good preference: the area under the receiver operating characteristic curves (AUROCs) for the cervical and endometrial cancer classifiers were 0.977 and 0.944, respectively, and the classifier for the origin of an adenocarcinoma yielded an AUROC of 0.939.   

In this paper, the proposed deep learning model is not clear. Although it is a black box, the description of the proposed method in the article is obviously insufficient. It is recommended to explain in large paragraphs (graphical explanations of the model and how the recognition of each layer changes, this should be explainable AI).  By the way, this paper should compare the performance of different deep learning models, and one deep learning approach Is Not enough (Only one kind of deep learning is used in this paper?). What kind of hardware environment is used for model calculation, and the software version is not clearly stated.

Author Response

The first author serves in the Department of Obstetrics and Gynecology, Seoul St. Mary's Hospital, College of Medicine. Uterine cervical and endometrial cancers have different subtypes with different clinical outcomes. I think the authors should have enough experience in processing relevant medical images.
-> JaeYen Song is an expert gynecologist and Soyoung Im is a pathologist specialized in gynecologic tumors. We carefully discussed the clinical needs in the analysis of gynecologic tumors to make the study clinically relevant. We published many papers on the deep learning-based analysis of pathologic tissues. We believe that we have enough experience in processing relevant medical images.  

In this paper, the proposed deep learning model is not clear. Although it is a black box, the description of the proposed method in the article is obviously insufficient. It is recommended to explain in large paragraphs (graphical explanations of the model and how the recognition of each layer changes, this should be explainable AI). 
-> We added a new Supplementary FigS1 to explain the structure of our DL model. Furthermore, the Materials and Methods section was modified for a more detailed explanation of the models. Because Inception v3 is a very complex model to be an explainable AI model, It is not easy to explain how the recognition of each layer changes during the training. 
(Line: 101-103)(Line: 136)(Line: 137-139)

By the way, this paper should compare the performance of different deep learning models, and one deep learning approach Is Not enough (Only one kind of deep learning is used in this paper?). 
-> As the reviewer suggested, it is of interest whether different deep learning models perform differently on the same task. In our early study for the discrimination of normal/tumor tissues [Ref. 15], we tested different deep learning models (AlexNet, ResNet-50, and Inception-v3) to compare their relative performance and the Inception-v3 yielded the best results. Furthermore, in our previous study on the subtype discrimination of gastric cancer tissues [Ref. 14], the Inception-v3 model showed good performance. Therefore, we decided to adopt the Inception-v3 model for the present study. We added description of why we adopted the Inception-v3 model in the Materials and Methods section. We think that the relative difference of performance between different tissue classification tasks can be easily compared because we have used the same Inception-v3 model for many studies.
(Line: 132-134)

What kind of hardware environment is used for model calculation, and the software version is not clearly stated.
-> We added a description of the hardware environment and software version in the Materials and Methods section.
(Line: 135)(Line: 142-144)

Reviewer 4 Report

Comments are attached.

Author Response

Thank you for the positive comments on our manuscript.

(1) In line 121: please explain why the 70 and 160 samples were selected, or explain why the rejected samples were not selected.
-> We explained the reason why 70 and 160 samples were selected in lines 84-87. When there are huge differences in the numbers of data between the classes, performance evaluation can be skewed by the majority class. Therefore, we randomly selected 70 and 160 patients for cervical squamous cell carcinoma and endometrioid endometrial adenocarcinoma, respectively, to make the differences between the major and minor classes under 1.5-fold. However, randomly sampled data from all training patients were used for the training of the classifiers to maximize the training efficacy as described in lines 123-127. The reason for the random selection is explained in the response to the next comment.

(2) In lines 123-127: when selecting images, have you considered that tumor image patches with different characteristics from the test sets should be selected instead of randomly selected?
-> We chose random selection rather than human selection to avoid selection biases from human selectors. Because deep learning can capture most discriminative features directly from data without feature extraction by human experts, we believe it is better to provide randomly selected massive data to avoid selection biases. A brief explanation was added in lines 129-130.

Round 2

Reviewer 1 Report

Accepted

Author Response

Thank you for the recommendation of acceptance of our manuscript.

Reviewer 3 Report

The authors have added some clear supplements to increase the readability of the paper, and this paper can be accepted and published after supplementary file(s) are added in the final version.

Author Response

Thank you for the recommendation of acceptance of our manuscript.

Supplementary files are added in the final version as suggested.